# Differentiation and Growth-Arrest-Related lncRNA (*DAGAR*): Initial Characterization in Human Smooth Muscle and Fibroblast Cells

**DOI:** 10.3390/ijms25179497

**Published:** 2024-08-31

**Authors:** Benjamin de la Cruz-Thea, Lautaro Natali, Hung Ho-Xuan, Astrid Bruckmann, Núria Coll-Bonfill, Nicholas Strieder, Víctor I. Peinado, Gunter Meister, Melina M. Musri

**Affiliations:** 1Mercedes and Martin Ferreyra Medical Research Institute, National Council for Scientific and Technical Research, National University of Córdoba (INIMEC-CONICET-UNC), Córdoba 5016, Argentina; bdlcruzthea@immf.uncor.edu (B.d.l.C.-T.); lnatali@immf.uncor.edu (L.N.); 2Regensburg Center for Biochemistry (RCB), Laboratory for RNA Biology, University of Regensburg, 93053 Regensburg, Germany; ho@med.uni-frankfurt.de (H.H.-X.); astrid.bruckmann@vkl.uni-regensburg.de (A.B.); 3Edward A. Doisy Department of Biochemistry and Molecular Biology, Saint Louis University School of Medicine, Saint Louis, MO 63104, USA; ncoll85@gmail.com; 4NGS-Core, LIT—Leibniz-Institute for Immunotherapy, 93053 Regensburg, Germany; nicholas.strieder@klinik.uni-regensburg.de; 5Department of Experimental Pathology, Institute of Biomedical Research of Barcelona (IIBB), CSIC, 08036 Barcelona, Spain; vpeinado@iibb.csic.es; 6Department of Pulmonary Medicine, Hospital Clínic, Biomedical Research Institut August Pi i Sunyer (IDIBAPS), University of Barcelona, 08036 Barcelona, Spain; 7Biomedical Research Networking Center in Respiratory Diseases (CIBERES), 28029 Madrid, Spain

**Keywords:** smooth muscle cells, long non-coding RNAs, *DAGAR*, cell cycle exit, fibroblasts, m6A

## Abstract

Vascular smooth muscle cells (SMCs) can transition between a quiescent contractile or “differentiated” phenotype and a “proliferative-dedifferentiated” phenotype in response to environmental cues, similar to what in occurs in the wound healing process observed in fibroblasts. When dysregulated, these processes contribute to the development of various lung and cardiovascular diseases such as Chronic Obstructive Pulmonary Disease (COPD). Long non-coding RNAs (lncRNAs) have emerged as key modulators of SMC differentiation and phenotypic changes. In this study, we examined the expression of lncRNAs in primary human pulmonary artery SMCs (hPASMCs) during cell-to-cell contact-induced SMC differentiation. We discovered a novel lncRNA, which we named Differentiation And Growth Arrest-Related lncRNA (*DAGAR*) that was significantly upregulated in the quiescent phenotype with respect to proliferative SMCs and in cell-cycle-arrested MRC5 lung fibroblasts. We demonstrated that *DAGAR* expression is essential for SMC quiescence and its knockdown hinders SMC differentiation. The treatment of quiescent SMCs with the pro-inflammatory cytokine Tumor Necrosis Factor (TNF), a known inducer of SMC dedifferentiation and proliferation, elicited *DAGAR* downregulation. Consistent with this, we observed diminished *DAGAR* expression in pulmonary arteries from COPD patients compared to non-smoker controls. Through pulldown experiments followed by mass spectrometry analysis, we identified several proteins that interact with *DAGAR* that are related to cell differentiation, the cell cycle, cytoskeleton organization, iron metabolism, and the N-6-Methyladenosine (m6A) machinery. In conclusion, our findings highlight *DAGAR* as a novel lncRNA that plays a crucial role in the regulation of cell proliferation and SMC differentiation. This paper underscores the potential significance of *DAGAR* in SMC and fibroblast physiology in health and disease.

## 1. Introduction

According to the World Health Organization, Chronic Obstructive Pulmonary Disease (COPD) is the third leading cause of death worldwide, with 3.23 million deaths in 2019 [1], and poses a significant socioeconomic burden [2]. COPD is characterized by chronic bronchitis with airway obstruction and lung emphysema [3]. While the primary affected tissues are the airways and the lung parenchyma, pulmonary vessel remodeling is a common occurrence facilitating the development of secondary pulmonary hypertension (PH) [4,5,6]. The presence of PH in COPD significantly impacts the quality of life and overall prognosis of COPD patients (reviewed in [7]). Pulmonary vessel remodeling involves the excessive proliferation of smooth muscle cells (SMCs) and fibroblasts, among other cell types within the vessel walls, resulting in increased pulmonary arterial pressure and pulmonary vascular resistance [8,9]. These phenomena have been previously related to sustained chronic inflammation, a common factor in the development of both COPD [10] and PH (Reviewed in [8]).

SMCs and fibroblasts are essential structural and functional components of the vessel wall. SMCs are located in the media of arteries and veins and contribute to blood flow regulation through the expression of a contractile smooth muscle phenotype. Specific transcription factors and contractility-associated proteins, such as myocardin (MYOCD), smooth muscle myosin heavy chain (MYH11), smooth muscle alpha actin (ACTA2 or αSMA), sm22α (TAGLN), and calponin1 (CNN1) are necessary for their function and are considered specific markers of the contractile SMC phenotype [11,12]. However, SMCs exhibit remarkable plasticity, allowing them to undergo phenotypic modulation in response to various stimuli and environmental cues [13]. This modulation involves a transition from a contractile, differentiated state to a synthetic, highly proliferative, dedifferentiated state [11]. In the latter, SMCs lose the expression of contractile SMC marker genes, experience increased proliferation and migration, and exhibit elevated synthesis of extracellular matrix components [11]. Understanding the mechanisms involved in the regulation of this cellular plasticity is crucial for the prevention and treatment of vascular diseases [13]. Fibroblasts, located primarily in the adventitia of arteries, respond similarly to SMCs to growth factors and inflammatory signals, promoting proliferation, migration, ACTA2 expression and extracellular matrix synthesis [14,15,16,17].

Non-coding RNAs have emerged as key regulators of gene expression and cell phenotype [18,19,20]. Long non-coding RNAs (lncRNAs), in particular, are non-protein-coding transcripts longer than 500 nucleotides, mostly transcribed by RNA polymerase II, and can exert diverse functions, especially depending on their localization and protein interactions (reviewed in [21]). In recent years, several lncRNAs involved in SMC homeostasis and phenotypic switching have been identified, mainly in the context of atherosclerosis (reviewed in [18]). For instance, *ANRIL* overexpression can inhibit the phenotypic switch and prevent atherosclerotic plaque development in vivo [22,23] while *SENCR* downregulation promotes proliferation and migration [24], SMILR expression induces proliferation through the regulation of the Centromere protein F (*CENPF*) mRNA [20,25] and *MYOSLID* enhances the vascular SMC differentiation program [26].

Recently, an old but still novel layer of regulation has been gaining increasing attention. RNA modification is emerging as a fundamental mechanism in cell homeostasis and cell fate decisions. In Eukaryotes, the most prevalent modification is N-6-Methyladenosine (m6A) [27]. This modification is involved in a variety of important cellular events such as mRNA decay [28,29], splicing [30] and translation [29], as well as in stem cell differentiation [31,32,33] and stress response [34,35]. In addition, it has been shown to be involved in the development of human abdominal aneurysm [36,37], pulmonary hypertension [38], and many other pathologies (reviewed in [39]). The m6A modification is deposited by a large enzymatic complex assembled around the methyltransferase METTL3 and its binding partner METTL14 [40]. Once methylation is set, reader proteins, such as the YTH domain-containing protein family, recognize and bind to the methyl group of the modified RNA to control its fate [28]. In particular, the canonical signaling for the m6A reader YTHDF2 involves the recruitment of the CCR4-NOT deadenylation complex and drives the P-Body localization of the modified RNA, which is then degraded [41]. Finally, the demethylases FTO and ALKBH5 can remove the methyl group [31,42,43], making this layer of regulation reversible and potentially very dynamic.

In the present work, we describe a novel lncRNA, which is expressed during human pulmonary artery SMC cell-to-cell contact-induced differentiation and in quiescent MRC5 lung fibroblasts. Its expression is associated with SMC differentiation and cell cycle arrest; hence, we named it Differentiation And Growth Arrest-Related lncRNA (*DAGAR*). In SMCs, at least two different *DAGAR* transcripts are expressed and upregulated during differentiation. In vitro, the treatment of SMCs with the pro-inflammatory cytokine Tumor Necrosis Factor alpha (TNF-α) modulates *DAGAR* expression. Notably, we found a decreased expression of *DAGAR* in total RNA derived from pulmonary arteries of COPD patients compared to control subjects, suggesting its regulation during disease development in vivo. Mass spectrometry analysis of proteins co-precipitated with *DAGAR* in RNA pulldown experiments, followed by Gene Ontology (GO) term enrichment analysis for biological processes, depicted associations with fundamental SMC differentiation and cell cycle exit-related processes. The association of *DAGAR* with the transferrin receptor (TFRC) further suggests a role for this non-coding RNA in iron metabolism. In addition, its association with the proliferation marker Ki67, together with the observation of increased Ki67 protein expression and enhanced proliferation following *DAGAR* downregulation, indicates a direct link to cell cycle regulation. Finally, *DAGAR* was enriched after m6A RNA pulldown and it was found to be regulated by the YTHDF2 reader, showing a pivotal role for *DAGAR* expression during cell cycle exit.

Overall, our findings provide the initial characterization of *DAGAR*, a novel lncRNA that plays a critical role in the modulation of the SMC phenotype and fibroblast cell cycle exit. We also present a comprehensive list of *DAGAR* protein interactors, shedding light on its function and underscoring its relevance in context-dependent signaling pathways associated with proliferation and differentiation. The regulation of *DAGAR* through pro-inflammatory signaling opens new avenues for the understanding and treatment of COPD and/or pulmonary hypertension.

## 2. Results

### 2.1. Expression of the lncRNA DAGAR Increases Significantly during Cell-to-Cell Contact-Induced Quiescence and Differentiation

To identify lncRNAs that undergo differential regulation during SMC phenotypic modulation, we employed an in vitro model of the cell-to-cell contact modulation of human pulmonary artery smooth muscle cells (hPASMCs) (Appendix A) as previously described by our [44,45] and other research groups [46,47]. An initial screening was performed by the total RNA isolation of proliferative (dedifferentiated) cells, referred to as D0, and confluent cells (representing the initial stage of differentiation), referred to as D2, for library preparation followed by RNA sequencing. Quality control and enrichment analysis of the differentially expressed genes at D2 with respect to D0 (Appendix A) showed the expected increase in the expression of genes related to the transforming growth factor beta (TGFβ) signaling pathway and elastic fiber formation, as well as a decrease in the expression of genes related to cell cycle progression (Appendix A). A prominent group of regulated genes was non-coding, of which 755 were classified as lncRNAs (Appendix A). To validate the differential expression of selected lncRNA candidates during SMC differentiation at D0 and D2 and in fully differentiated SMCs (D6), as well as their basal expression in endothelial cells (EC), RT-qPCR was performed (Appendix A). Gene expression changes of the candidates were successfully confirmed and the lncRNA4, which we now refer to as *DAGAR* (TCONS_00006193, CATG000061625.1, ENSG00000288022), was selected for further characterization. Selection was based on its expression profile, which was highest in differentiated SMCs and nearly undetectable in proliferative SMCs or confluent endothelial cells (Appendix A). *DAGAR* is encoded on chromosome 3, at band 3p21.2, located between the *HEG1* and *SLC12A8* genes (Figure 1A). The FANTOM CAT expression database [48] has annotated six transcript variants of different lengths (194 nt, 2938 nt, 3173 nt, 3386 nt, 3452 nt, and 3674 nt) while the Genome Browser database (hg 38) only annotates one variant (1481 nt).

Specific variants expressed in SMCs were identified by RT-PCR using specific primers targeting distinct regions, cloning in the pGEMT-easy vector, and subsequent Sanger sequencing (Figure 1A). Cloning was successfully achieved in the region with intronic variability (at the transcript’s beginning) and in the middle region of the transcript identifying MICT00000249499.1 and MICT00000249498.1, designated as *DAGAR-1* and *DAGAR-2*, respectively. No fragments corresponding to the other variants were detected in SMCs. RT-qPCR analysis, using primers capable of detecting all the variants, exhibited a significant upregulation of *DAGAR* in confluent respect to proliferative SMCs (Figure 1B). Similar fold changes for *DAGAR-1* and *DAGAR-2* were observed in these conditions (Appendix A). Northern blot analysis, using two different probes, both able to target all isoforms, validated the increase in *DAGAR* expression during SMC differentiation (Figure 1C). Total polyadenylated RNA analysis after oligo(dT) pulldown followed by RT-qPCR confirmed the presence of polyadenylated mature *DAGAR* transcripts (Appendix A).

Taken together, these results indicate a strong upregulation of at least two variants of *DAGAR*, a novel polyadenylated lncRNA, at the onset of quiescence in SMCs when compared to proliferative cells.

In addition, the expression of *DAGAR* was also observed in fibroblasts. Notably, *DAGAR* expression significantly increased in MRC5 fibroblast cells upon cell cycle exit induced either by cell-to-cell contact (Figure 1D), 48 h of serum starvation (Figure 1D), or 24 h of cell cycle inhibition using the reversible cyclin-dependant-kinase-1 (CDK1) inhibitor Ro3306 (Figure 1D) compared to their respective control conditions. Importantly, *DAGAR* expression returned to basal levels after 24 h of CDK1 inhibitor withdrawal (Figure 1D), indicating an inverse relationship between *DAGAR* expression and proliferation. These data demonstrate that *DAGAR* is not restricted to SMCs and that it is upregulated during cell cycle exit.

### 2.2. DAGAR Expression Is Decreased in Pro-Inflammatory Conditions

The cytokine TNFα has been recognized as an inducer of SMC phenotypic modulation [49,50] and is involved in the inflammatory pathway activating the proliferation of SMCs and fibroblasts in both COPD and PH. Taking this into consideration, we tested whether treatment with TNFα modulates *DAGAR* expression. Remarkably, we found that a 48 h treatment with TNFα was sufficient to downregulate *DAGAR* expression in SMCs at D6 to approximately half of the control levels (Figure 2A). Given *DAGAR* modulation after TNFα treatment, and a similar regulation for SMCs and fibroblasts during quiescence, we wondered whether COPD patients would display a similar change. To this end, we measured its expression in total RNA extracted from human pulmonary arteries of patients with COPD compared to non-smoking control patients (NS). This retrospective analysis was performed on samples from the study conducted by Musri et al. [45] and employed RT-qPCR for characterization. Our findings showed a significant downregulation of *DAGAR* in pulmonary arteries from COPD patients when compared to non-smoker patients (NS) (Figure 2B), similar in relative change to that observed after TNFα treatment on SMCs. These results suggest a potential correlation between the downregulation of *DAGAR* and the pathological states of SMCs and/or adventitial fibroblasts in these patients.

### 2.3. DAGAR Knockdown Promotes SMC Proliferation and Hinders mRNA Expression of Important SMC Genes

Following the observation of the induction of *DAGAR* during cell cycle exit, we next explored its involvement in cell cycle regulation. To this end, we employed specific siPOOLs (siDAGAR, targeting all isoforms) to downregulate *DAGAR* expression in SMCs. The knockdown efficiency for *DAGAR-1* and *DAGAR-2* analyzed by RT-qPCR reached approximately 60–70% compared to the control condition (Appendix A). *DAGAR* knockdown in polarized cells (D2) resulted in an increase in the percentage of cells in the S phase of the cell cycle observed by flow cytometry (Figure 2C). Additionally, *DAGAR* knockdown at D2 led to decreased mRNA levels of *MYOCD*, *CNN1*, and *TAGLN* (Figure 2D).

These findings indicate that *DAGAR* not only exhibits induction during cell cycle exit but is also important in order to complete this process and demonstrates a key role for this lncRNA in the induction of master genes that belong to the SMC contractile phenotype at the mRNA level.

### 2.4. DAGAR Interactors Are Involved in Key SMC-Related Pathways

Next, to identify its interaction partners, we performed RNA affinity purification by specifically pulling down endogenous *DAGAR* expressed in confluent SMCs using biotinylated probes antisense to the transcript followed by mass spectrometry analysis (DAGAR-RIP). After excluding proteins obtained in control conditions (Figure 3A), data analysis resulted in a list of 240 proteins that were specifically bound to *DAGAR* (Appendix A).

To better understand the molecular pathways in which *DAGAR* is involved, we investigated the functions of *DAGAR*-interacting proteins. We conducted the Gene Ontology (GO) biological-process term enrichment analysis of the precipitated proteins set (Appendix A) and Reactome Pathway enrichment analysis (Appendix A). Additionally, we employed the Revigo clustering of significantly enriched GO terms (adjusted *p* value < 0.05) to visualize a non-redundant set of biological-process GO terms [51]. This unbiased analysis grouped the terms into prominent clusters and small groups of GO terms (Figure 3B). Main clusters were associated with multiple biological processes including cell differentiation, developmental growth, and morphogenesis (cluster 1 and 2), cell cycle regulation (cluster 3), and ncRNA metabolism (clusters 7 and 8). The identification of such interactors provides valuable insights into the mechanisms underlying the regulatory functions of *DAGAR* in SMCs.

As a means of *DAGAR*–bound protein cross-validation, the association of *DAGAR* with transferrin receptor 1 (TRFC) was investigated. TFRC is the most important receptor regulating iron metabolism, as well as regulating SMC proliferation and vascular remodeling [52]. Therefore, we conducted the immunoprecipitation of GPF (IP-GFP) from total lysates of confluent MRC5 previously transfected with plasmids encoding for either GFP-TFRC or GFP alone for 72 h plus 24 h of treatment with 10 µM Ro3306 to induce maximum *DAGAR* expression. After RNA isolation from the IP-GFP, followed by RT-qPCR, we were able to detect the presence of *DAGAR* in the GFP-tagged TFRC but not in the GFP control condition indicating the specific binding of *DAGAR* with TFRC (Figure 3C). These data suggest a potential role for *DAGAR* in iron homeostasis.

### 2.5. DAGAR Knockdown Is Associated with Impaired SMC Contractile Cytoskeletal Formation and Increased ki67 Nuclear Protein Expression

A noteworthy observation was that the top enriched Reactome pathway for *DAGAR* interactors was “Rho-GTPase regulation” (Appendix A), a pathway that has been shown to play a crucial role in cytoskeleton dynamics [53,54]. This aligns with the Biological Processes observed in clusters 4a and 4b of significantly enriched GO terms (Figure 3B) Given the importance of cytoskeletal assembly for proper contractility function in smooth muscle, we assessed the impact of *DAGAR* knockdown on SMC contractile cytoskeleton maturation. We analyzed ACTA2 and CNN1 fiber formation by immunofluorescence and found that *DAGAR* knockdown hinders ACTA2 (Figure 4A) and CNN1 assembly (Figure 4B). Additionally, in our RNA-IP mass spec data, we found a direct interaction of *DAGAR* with the proliferation marker Ki67. To address this interesting observation, and taking into account that *DAGAR* silencing promotes an increase in the percentage of cells on the S phase, we analyzed Ki67 expression after the silencing of *DAGAR* by immunofluorescence and qPCR (Figure 4C–E). Surprisingly, the observed phenotype was a significant increase in the percentage of Ki67-positive nuclei after knockdown despite a reduction in *Ki67* mRNA. Although this observation requires further investigation, the data suggest that the interaction of *DAGAR* with Ki67 leads to its protein downregulation at the translational or post-translational level.

### 2.6. DAGAR Interactors Shed Light on Its Regulation by the N-6-Methyladenosine (m6A) Reader YTHDF2

From the co-precipitated proteins obtained in the RNA-IP experiments, CNOT1 and VIRMA were found to be bound to *DAGAR* (Appendix A). Therefore, this hinted at a possible mechanism of regulation for *DAGAR* expression. CNOT1 is a constitutive part of the CCR4-NOT deadenylation complex, the final effector of the m6A RNA decay machinery [41]. Since this complex is recruited by the m6A reader YTHDF2 to exert its canonical function, and given that VIRMA is a constitutive part of the m6A methyltransferase complex, we sought to investigate the relation of *DAGAR* to this pathway.

To this end, we first corroborated the presence of m6A modification within the *DAGAR* transcript by m6A immunoprecipitation, followed by RT-qPCR. The enrichment of *DAGAR* observed in m6A-IP over IgG-IP confirmed the presence of this modification within the transcript (Figure 5A). Next, we silenced the canonical reader for mRNA decay YTHDF2 in both SMCs (Appendix A) and MRC5 (Appendix A) cells and observed significant increases in the expression of *DAGAR*, similar for both cell types (Figure 5B,C, respectively). Moreover, the expression of YTHDF2 in quiescent MRC5 cells was diminished when compared to proliferative cells in a similar way to that observed after cell cycle blockade by 24 h treatment of Ro3306 in proliferative MRC5 cells (Figure 5D). Since YTHDF2 is downregulated after Ro3306 treatment in other cell lines in a proteasome-dependent way [55], we treated quiescent MRC5 cells with MG132, a well-characterized proteasome inhibitor, and Brefeldin A to test the alternative of non-canonical autophagy involvement in this regulation. Only proteasome inhibition had an effect on YTHDF2; in fact, 4 h of MG132 treatment was sufficient to stabilize the YTHDF2 protein in quiescent MRC5 cells and showed a strong *DAGAR* downregulation, mimicking *DAGAR* expression in proliferative MRC5 cells (Figure 5E). These results indicate that YTHDF2 protein regulation in quiescent MRC5 cells occurs via a proteasome-dependent pathway and that *DAGAR* is regulated by the m6A machinery.

## 3. Discussion

The findings presented herein for human smooth muscle cells (SMCs) were obtained in the context of an in vitro model previously characterized and used by our lab and others [44,45,46,47]. Briefly, the model consists of cell-to-cell contact-dependent cell cycle arrest and SMC differentiation, as denoted by the increased mRNA expression of key transcription factors and co-factors (*GATA6*, *MYOCD*), and molecules involved in elastic fiber formation and contractility (*ACTA2*, *TAGLN*, *CNN1*, *MYH11*) [44]. This is not limited to mRNA expression since the assembly of CNN1, ACTA2, and TAGLN fiber formation has been observed and characterized [44]. This model consists of three main time points for evaluation, involving proliferative cells (D0), which were hPASMCs plated at 70% confluency and harvested the next day; a second time point 48 h posterior to the proliferative cell state, when they attained full plate confluency and exhibited polarizing behavior (D2); and 96 h after full confluency (D6), when they exhibited the maximum expression of contractility-associated mRNAs and proteins within the model. The main purpose of using this cell model was based on two key points. The first was that the molecules discovered could be directly relevant and translatable to human health and disease, given their human origin, and the second was the use of a differentiation model that did not rely on exogenous cytokine administration, rendering it closer to a non-pathological wound healing process for these cells, often occurring after exposure to harmful events. To study the phenomena related to dedifferentiation cues, the pro-inflammatory cytokine TNFα was used, modeling the effect of an inflammation-mediated environment dysregulation in differentiated SMCs [44], a cue commonly found in pathologies that include a vascular remodeling component such as atherosclerosis, pulmonary hypertension, COPD, and obesity [49,50]. Despite the overall limitations of in vitro models, there are several advantages for the purpose of new molecule discovery. Nevertheless, further research will be needed to assess the effects of the in vivo perturbation of such candidates and their applicability to human disease diagnosis and/or treatment.

The results of this study reveal a novel human long non-coding RNA, herein referred to as “*DAGAR*” (TCONS_00006193), which exhibits a strong association with SMC differentiation and cell cycle exit. Two variants were identified by cloning and sequencing experiments: MICT00000249498.1 (*DAGAR-1*), spanning 3,674 bases, and MICT00000249499.1 (*DAGAR-2*), spanning 2938 bases, as annotated in the FANTOM CAT expression database. Consistently, the Northern blot signal that showed a significant upregulation at D2 and D6 with respect to D0 was detected within the corresponding size range of 3 and 4 Kb. Of note, the expression of *DAGAR* in proliferative SMCs (D0) was nearly null, ramping up quickly at full plate confluency; however, even at the highest differentiation point, the absolute abundance of this transcript was very low, hindering a clear distinction of individual bands. Nevertheless, a blurred double band was observed at sizes corresponding to *DAGAR-1* and *DAGAR-2*. Although we did not detect other variants in SMCs, the production of the 3,173 Kb MICT00000249500.1 cannot be ruled out. In line with the Northern blot data, *DAGAR* exhibited significant upregulation during SMC differentiation when targeted by primers capable of amplifying all variants, with minimal detectable expression in proliferative SMCs. The RT-qPCR amplification cycle (Cq), when using variant-specific primers, was higher for *DAGAR-2,* indicative of a lower basal expression of this transcript. Despite this difference, the observed fold change was similar for *DAGAR-1* and *DAGAR-2*, and therefore, in subsequent studies, we analyzed *DAGAR* expression using general primers able to amplify all detected variants. We also observed *DAGAR* induction in fibroblasts following cell cycle arrest induced by different means, including cell-to-cell contact, serum starvation, and CDK1 inhibition. *DAGAR* knockdown in SMCs at D2 not only induced SMC proliferation but also hindered the acquisition of a SMC contractility-associated transcriptional plan, including *MYOCD*, *CNN1*, and *TAGLN* mRNA expression, as well as ACTA2 and CNN1 fiber formation at the protein level. These findings demonstrate the essential role of *DAGAR* in the acquisition and/or maintenance of a contractile-differentiated SMC phenotype. The proteins found to interact with DAGAR support the interpretation that the failure of ACTA2 and CNN1 fiber formation after DAGAR knockdown could be related to Rho-GTPase regulation. In this regard, future experiments will shed light into specific Rho-GTPase effector regulation and specific pathways involved.

Indicative of a possible role in vivo, *DAGAR* expression was found to be decreased in pulmonary arteries from COPD patients. COPD is characterized by airflow obstruction resulting from an inflammatory process [56] and vascular remodeling in COPD involves the excessive proliferation and dedifferentiation of both SMCs as well as adventitial fibroblasts [11,56,57]. Despite the number of analyzed patients in our study being limited to retrospective existences, these data support the correlation between *DAGAR* expression and the contractile-quiescent state of SMCs in pulmonary arteries in vivo. This finding leads us to interpret that the downregulation of *DAGAR* may directly contribute to SMC proliferation and phenotypic switching while, at the same time, being indicative of proliferative disorders of SMCs and/or fibroblasts. This could open new avenues for the rapid diagnosis of chronic inflammation-associated pathologies. Nevertheless, further experiments assessing *DAGAR* regulation in other types of fibroblasts (i.e., dermal fibroblasts) during chronic inflammation in humans need to be evaluated to identify whether *DAGAR* expression can be used as a biomarker for such purposes. TNFα, a cytokine involved in the pathogenesis of pulmonary vascular remodeling [10], is well known for its ability to induce SMC proliferation, dedifferentiation, and calcification [49,50]. A previous study by Nick Morrell’s laboratory revealed that TNFα promotes pulmonary hypertension by repressing BMPR-II signaling and enhancing BMP6, which, in turn, stimulates SMC proliferation via ALK2 and ACTR-IIA [58]. In this study, we observed a 50% reduction in *DAGAR* expression in SMCs incubated with TNFα for 48 h compared to control cells. Interestingly, TNFα post-transcriptionally promotes the expression of TFRC and other iron-import-related proteins, leading to iron accumulation in the endothelium [59] and accelerating iron-induced SMC calcification [60]. In this regard, our endogenous *DAGAR* pulldown followed by mass spectrometry revealed a significant enrichment of proteins related to iron metabolism and transferrin transport (cluster 6 of Figure 3B), including TFRC, the Myotonic-dystrophy-kinase-related Cdc42-binding kinase α (CDC42BPA), and Heat Shock Protein Family A (Hsp70) Member 9 (HSPA9) (Appendix A). Iron is a fundamental micronutrient required for a variety of cellular processes including metabolism, respiration, and DNA synthesis; hence, it is also required for proliferation [61]. TFRC is the most crucial membrane-bound receptor that mediates iron uptake through the clathrin-mediated endocytosis of iron-loaded transferrin, the major iron transport protein in the blood [61]. TFRC protein is also involved in the regulation of proliferation [61,62], as well as in the development of diseases such as cancer [63], atherosclerosis [60], pulmonary vascular remodeling [52], asthma [56], and lung fibrosis [64]. CDC42BPA is a serine/threonine protein kinase that colocalizes and interacts with TFRC-loaded transferrin, regulating its uptake [65]. In addition, CDC42BPA is a downstream effector of the small GTPase CDC42, which was among the top enriched Reactome pathways identified from DAGAR interactors. Heat Shock Protein Family A (Hsp70) Member 9 (HSPA9) participates in the mitochondrial Fe-S cluster biogenesis and has been reported to regulate proliferation [66,67]. The association of *DAGAR* with TFRC suggests a role for *DAGAR* in iron metabolism. Further experiments have been planned to understand the significance of *DAGAR* interaction with TFRC and related proteins, as well as its impact on iron homeostasis.

Besides the aforementioned, the unbiased analysis of *DAGAR* protein partners showed the enrichment of Gene Ontology biological-processes term clustering around key cellular processes associated with cell differentiation and cell cycle exit (clusters 2 and 3 of Figure 3B). Consistent with these findings, DAGAR silencing resulted in an increased percentage of cells in the S phase and enhanced the nuclear localization of Ki67. Ki67 was co-precipitated with *DAGAR*, hinting at negative post-translational regulation, since its mRNA did not follow the same pattern of expression after *DAGAR* silencing. However, this observation requires further exploration.

Finally, the association of *DAGAR* with several proteins related to ncRNA, and RNA metabolism (clusters 7 and 8 of Figure 3B)*,* such as members of the N-6-methyladenosine (m6A) methylation machinery and mediators of RNA-induced gene silencing and microRNA pathways, indicates potential regulatory mechanisms affecting *DAGAR* expression. In this regard, the association of *DAGAR* with CNOT1, the core component of the CCR4-NOT deadenylation complex, and with VIRMA, the core protein of the METTL3-METTL14 methyltransferase complex, hinted at a regulatory mechanism for *DAGAR* by m6A modification and YTHDF2-mediated recognition and decay. The enrichment of *DAGAR* after m6A immunoprecipitation strengthened the interpretation that such interactions were relevant for *DAGAR* regulation. Further experiments revealed that the knockdown of YTHDF2 was enough to increase three-fold *DAGAR* expression. Experiments blocking cell cycle progression using Ro3306 in MRC5 cells induced *DAGAR* expression, similar to the induction observed when cells reached confluency. Closely related, YTHDF2 expression has been previously related to cell proliferation, working as a mediator in cell cycle progression, and has been demonstrated to be downregulated after CDK1-inhibition-mediated cell cycle exit by Ro3306 treatment in Hela cells [55] in a proteasome-dependent fashion. Proteasome blocking by MG132 is able to restore YTHDF2 in confluent MRC5 cells, supporting the notion that YTHDF2 regulation at cell confluency in MRC5 is similar to that observed after Ro3306-induced cell cycle exit in HeLa cells while, at the same time, downregulating *DAGAR* RNA levels, arguing in favor of a proteasome-dependent mechanism of control for upstream regulators of this lncRNA. It is worthwhile to note that YTHDF2 downregulation has been proven necessary for pluripotent stem cell commitment to specific lineages [33] and has been recently reported to induce SMC proliferation, participating in the development of vascular remodeling during PH through *MYADM* transcript stabilization [68]. These observations were in line with the findings reported herein, with the difference that the regulation mechanism proposed for *DAGAR* was through the canonical mRNA decay pathway of the YTHDF2 reader.

In summary, this paper documents a novel lncRNA that we have named *DAGAR* due to both its strong association with both SMC and fibroblast cell cycle exit as well as its requirement for proper SMC differentiation. Our findings demonstrate that the treatment of differentiated SMCs with TNFα, a known inducer of SMC dedifferentiation and proliferation, leads to *DAGAR* downregulation. Conversely, the knockdown of *DAGAR* promotes SMC proliferation. These results are in line with the decreased *DAGAR* expression observed in the pulmonary arteries of COPD patients. Therefore, TNFα-induced SMC dedifferentiation may occur, at least in part, through the downregulation of *DAGAR* expression. The proteins found to interact with *DAGAR* are a foundational stone from where further research will be conducted. The findings presented herein relate *DAGAR* with numerous biological processes, ranging from protein stability to cytoskeleton regulation, and present a strong regulatory mechanism hypothesis for *DAGAR* through the m6A/YTHDF2/CCR4-NOT axis during cell cycle exit in a proteasome-dependent pathway. Collectively, our data unveil *DAGAR* as both a new lncRNA needed for the establishment and maintenance of a quiescent-differentiated SMC phenotype and cell cycle exit for SMCs and fibroblasts, as well as a previously unknown player in inflammatory signaling pathways, highlighting a prospective therapeutic target for vascular and fibrotic diseases and a potential biomarker for chronic inflammation-related diseases.

## 4. Materials and Methods

### 4.1. Cell Culture

Human pulmonary artery smooth muscle cells (HPASMCs, Lonza Group, Basel, Switzerland) were cultured as in previous studies [44,45] using SM-bullet Medium (Lonza Group, Switzerland) supplemented with 10% fetal bovine serum (FBS. Gibco, Thermo Fisher, Waltham, MA, USA). Cells were used between passages 3 and 8. The cell-to-cell contact-induced differentiation model, which had been previously used by our laboratory [44,45], was employed. Briefly, hPASMCs were plated at 70% confluence and harvested 24 h later, designated as D0 (proliferative cells). D2 represented cells at full confluency (at the beginning of differentiation) and D6 corresponded to cells harvested 96 h after reaching full confluency (fully differentiated SMCs). Medium was changed every 48 h. Human pulmonary artery endothelial cells (hPAECs, Lonza Group, Switzerland) were cultured according to manufacturer’s instructions and supplemented with 10% FBS (Gibco, Thermo Fisher, USA). Human embryonic kidney 293 cells (ATCC, Manassas, VA, USA) and human fetal lung fibroblasts MRC5 (ATCC, USA) were cultured in Dulbecco’s Modified Eagle Medium (DMEM) (Sigma-Aldrich, Merck KGaA, St. Louis, MO, USA) supplemented with 10% FBS (Gibco, Thermo Fisher, USA) and 1% Penicillin/Streptomycin antibiotic mix (Sigma-Aldrich, USA). All cells were cultured under standard conditions in cell culture incubators with 95% humidity and 5% CO_2_. Additionally, they were tested for contamination with Mycoplasma spp.

### 4.2. Cell Treatments

Dedifferentiation of SMCs was achieved by treating differentiated (D6) cells with 10 ηg/mL of TNFα (R&D Systems, Minneapolis, MN, USA) for 48 h as previously described [44].

Cell cycle arrest was induced in proliferative MRC5 fibroblasts with the ATP-competitive CDK1 reversible inhibitor Ro3306 at a final concentration of 10 µM (Sigma-Aldrich, Merck KGaA, USA) for 24 h in growth medium as reported in previous studies [69]. Treatment with the same volume of vehicle DMSO for 24 h was used as vehicle control.

Proteasome activity was inhibited in confluent MRC5 cells for 4 h with the proteasome inhibitor MG-132 at a final concentration of 20 μM (Sigma-Aldrich, Merck KGaA, USA) in DMEM medium (Sigma-Aldrich, Merck KGaA, USA) supplemented with 10% FBS (Gibco, Thermo Fisher, USA). Treatment with the same volume of DMSO for 4 h was used as vehicle control. Brefeldin A (Sigma-Aldrich, Merck KGaA, USA) treatment at 10 μg/μL for 4 h in DMEM medium (Sigma-Aldrich, Merck KGaA, USA) supplemented with 10% FBS (Gibco, Thermo Fisher, USA) was used to evaluate non-canonical autophagy.

### 4.3. mRNA from Human Pulmonary Arteries

We conducted a retrospective study using pulmonary artery (PA) segments obtained from patients undergoing surgical lung resection for localized lung neoplasms over a period of two years [45]. The study included mRNA from 5 non-smoker control patients (NS) and 8 patients diagnosed with COPD. Ethical approval for the study was obtained from the Ethic Committee of the Hospital Clinic, Barcelona, Spain.

### 4.4. Gene Silencing Using siPOOL siRNAs

siPools (siTOOLS^®^, Germany [70]) targeting *DAGAR* (20 or 40 ηM) or YTHDF2 (3 ηM) were reverse-transfected using serum-free OptiMEM (Gibco, Thermo Fisher, USA) and RNAiMAX lipofectamine (Invitrogen, Thermo Fisher, West Grove, PA, USA) according to the manufacturer’s instructions. As a transfection control, siPool directed against scrambled sequences (siCTs) was used at the same concentration as the specific target-directed siPools.

### 4.5. Immunofluorescence

Cells were seeded in 24-multiwell plates over 0.2% gelatin-coated coverslips and cultured under the conditions described previously in the Cell Culture section. Fixation was carried out for 20 min at room temperature (RT) using a fixation solution (4% Paraformaldehyde in PBS, pH 7.4). Then, permeabilization solution (3% bovine serum albumin (BSA), 0.2% Triton X-100 in PBS) was added and incubated for 10 min at RT. After each step, coverslips were washed twice with wash solution (0.2% BSA and 0.1% Tween 20 in PBS). For primary antibody incubation, coverslips were placed in a humid chamber and incubated overnight at 4 °C in wash solution. Antibodies used were anti-α-SMA and anti-CNN1 from DAKO (Cytomation, Carpinteria, CA, USA) and anti-Ki67 from Novocastra (Newcastle, UK). Incubation with secondary Alexa Fluor antibodies (Invitrogen, Thermo Fisher, West Grove, PA, USA) was performed in wash solution for 1 h at RT. Finally, coverslips were mounted using ProLong Gold (Thermo Fisher, USA), which included DAPI labeling.

### 4.6. RNA Extraction and Real-Time Quantitative PCR (RT-qPCR)

Total RNA for Northern blot, immunoprecipitation, and ribonucleoprotein affinity purification experiments was extracted using TRIzol (Invitrogen, Thermo Fisher, USA) according to the manufacturer’s instructions. For qPCR experiments, RNA isolation was performed using the NucleoSpin RNA Column extraction kit (Macherey-Nagel, GmbH & Co. KG, Düren, Germany). The concentration of total RNA was measured using NanoDrop 2000 (NanoDrop Products, Wilmington, DE, USA). Reverse-transcription reaction was performed using 1 µg of total RNA with the ReverAid First Strand cDNA Synthesis Kit (Thermo Fisher, USA). Quantitative real-time PCR was conducted using SYBR GREEN (either SsoFast EvaGreen supermix, Bio-Rad Laboratories, Hercules, CA, USA or Takyon EUROGENTEC, Seraing, Belgium) by using the CFx96 qPCR equipment (Bio-Rad Laboratories, USA). Specific primers used are listed in Appendix A. Relative changes in gene expression were calculated using the double delta Ct method (2^−ΔΔCt^) and normalized to Glyceraldehyde 3-phosphate dehydrogenase (GAPDH).

### 4.7. RNA Sequencing

A total of 1 μg of RNA extracted from proliferative SMC (D0) and confluent cells at the beginning of differentiation (D2) was used to generate libraries using TrueSeq RNA Preparation Kit (Illumina, San Diego, CA, USA). The libraries were prepared following the manufacturer’s indications. Then, the libraries were sequenced using a HiSeq 2000 platform (Illumina, USA). The data have been submitted to the GEO database (accession number GSE242196).

### 4.8. Bioinformatic Analysis

Data normalization, control, and enrichment calculations were performed using R studio [71] with the following packages: dplyr v1.1.4 [72], ggplot2 v3.5.0 [73], EnsDb.Hsapiens.v86 v2.99.0 [74], DOSE v3.82.2 [75], clusterProfiler v4.10.1 [76], ReactomePA v1.46.0 [77], DESeq2 v1.42.1 [78], and UniprotR v2.4.0 [79]. To ensure proper visualization of color scales in the Reactome pathway enrichment analysis graph performed in R v4.3.2, the colorblind-friendly mapping Viridis v0.6.5 [71] was used.

### 4.9. RNA Affinity Purification

Differentiated SMCs (D6) were crosslinked by incubation with 4% formaldehyde in PBS for 10 min at RT to preserve transient RNA–protein interactions. The reaction was then quenched by adding 1% glycine in PBS for 5 min. Cells were then lysed in 1 mL of lysis buffer (25 mM Tris-HCl pH 7.5, 150 mM KCl, 2 mM EDTA, 1 mM NaF, 0.5% NP-40, 1 mM DTT and 500 µM AEBSF, 40 U/mL Ribolock in milliQ water). To obtain nucleic acid fragments of approximately 300-600 nucleotides in length, the lysate was sonicated for 20 min at 4–8 °C in a Covaris S220 sonicator (Covaris Inc., Woburn, MA, USA). Specific raPOOLs designed to target DAGAR were used according to manufacturer instructions. This approach had been previously used for LINC00152 [80] (siTOOLS, Martinsried, Germany). Briefly, the cell lysate was incubated with a pool of biotinylated DNA probes designed for targeting *DAGAR*. Pulldown was performed using streptavidin-coated magnetic beads (Invitrogen). Co-precipitated proteins were extracted, subjected to gel electrophoresis, and then silver stained. Specific bands of interest were excised, purified, and sent for mass spectrometry analysis. To ensure specificity, two negative controls were used: unspecific probes targeting LacZ and specific probes targeting *DAGAR* in a cell line lacking *DAGAR* expression (HEK293 cells). By eliminating nonspecific targets using both negative controls, we obtained a list of potential binding partners of *DAGAR*.

### 4.10. Mass Spectrometry

For mass spectrometry analysis of proteins, slices were cut out from a silver-stained gel, transferred into 2 mL micro tubes (Eppendorf, Hamburg, Germany), and destained by incubating with 15 mM K3Fe (CN)6/50 mM Na_2_S_2_. Subsequently, gel slices were washed with 50 mM NH_4_HCO_3_, 50 mM NH_4_HCO_3_/acetonitrile (3/1), and 50 mM NH_4_HCO_3_/acetonitrile (1/1) while being shaken gently in an orbital shaker (VXR basic Vibrax, IKA, Barcelona, Spain). Gel pieces were lyophilized after shrinking by 100% acetonitrile. To block cysteines, reduction with DTT was carried out for 30 min at 57 °C followed by an alkylation step with iodoacetamide for 30 min at room temperature in the dark. Subsequently, gel slices were washed and lyophilized again as described above. Proteins were subjected to in-gel tryptic digestion overnight at 37 °C with approximately 2 µg trypsin per 100 µL gel volume (Trypsin Gold, mass-spectrometry-grade, Promega, Madison, WI, USA). Peptides were eluted twice with 100 mM NH_4_HCO_3_ followed by an additional extraction with 50 mM NH_4_HCO_3_ in 50% acetonitrile. Prior to LC-MS/MS analysis, combined eluates were lyophilized and reconstituted in 20 µL of 1% formic acid. Separation of peptides by reversed-phase chromatography was carried out on an UltiMate 3000 RSLCnano System (Thermo Scientific, Dreieich, Germany) that was equipped with a C18 Acclaim Pepmap100 preconcentration column (100 µm i.d. × 20 mm, Thermo Fisher, USA) in front of an Acclaim Pepmap100 C18 nano column (75 µm i.d. × 150 mm, Thermo Fisher, USA). A linear gradient of 4% to 40% acetonitrile in 0.1% formic acid over 90 min was used to separate peptides at a flow rate of 300 nL/min. The LC-system was coupled on-line to a maXis plus UHR-QTOF System (Bruker Daltonics, Bremen, Germany) via a CaptiveSpray nanoflow electrospray source (Bruker Daltonics, Bremen, Germany). Data-dependent acquisition of MS/MS spectra by CID fragmentation was performed at a minimum resolution of 60,000 for MS and MS/MS scans. The MS spectrum rate of the precursor scan was 2 Hz, processing a mass range between *m*/*z* 175 and *m*/*z* 2000. Via the Compass 1.7 acquisition and processing software (Bruker Daltonics, Bremen, Germany), a dynamic method with a fixed cycle time of 3 s and an *m*/*z*-dependent collision energy adjustment between 34 and 55 eV was applied. Raw data processing was performed in Data Analysis 4.2 (Bruker Daltonics, Bremen, Germany), and Protein Scape 3.1.3 (Bruker Daltonics, Bremen, Germany) in connection with Mascot 2.5.1 (Matrix Science, Mount Prospect, IL, USA) facilitated database searching of the Swiss-Prot Homo sapiens database (release-2020_01, 220420 entries). Search parameters were as follows: enzyme specificity trypsin with 2 missed cleavages allowed, precursor tolerance 0.02 Da, MS/MS tolerance of 0.04 Da, carbamidomethylation or propionamide modification of cysteine, oxidation of methionine, deamidation of asparagine and glutamine, and acetylation of protein N-termini; these were set as variable modifications. Mascot peptide ion-score cut-off was set at 30. Protein list compilation was completed using the Protein Extractor function of Protein Scape.

### 4.11. Protein Interaction Cross-Validation

1 × 10^6^ MRC5 fibroblasts were transfected with 3 µg of either pLL3.7 expressing GFP protein or a plasmid expressing TfR-GFP fusion protein [81] by electroporation using the AMAXA Nucleofactor 2 system (Lonza Group, Switzerland). Medium was changed after 24 h and plasmid was expressed for 72 h. Then, cells were treated with 10 µM Ro3306 for 24 h to trigger more *DAGAR* expression. Cells were harvested by mechanical detachment using a cell scraper and washed twice with ice-cold PBS. A total of 200 µL of lysis buffer (25 mM Tris HCl pH 7.5, 150 mM KCl, 2 mM EDTA, 1 mM NaF, and 0,5% NP-40 supplemented with freshly added AEBSF 1 mM (1:100), DTT 1 mM (1:1000), and 1:1000 Ribolock (40 U/µL)) was added per sample and incubated on ice for 30 min with occasional mixing. Samples were then centrifuged at 17,000 rcf for 20 min and the supernatant was recovered. The volume was adjusted to 300 µL with lysis buffer. From these, 10% was taken as the input and 90% was used for immunoprecipitation (IP). The lysate was incubated with 12 µg of anti-GFP antibody (Abcam, Waltham, MA, USA, chicken polyclonal, ab13970) previously coupled to 50 µL of Protein G Agarose beads (GE Healthcare, Chicago, IL, USA) per IP. Total volume was adjusted to 1.3 mL with lysis buffer and the samples were incubated for 3 h in rotation at 4 °C. Subsequently, the beads were spun down (1000 rfc for 2 min at 4 °C) and washed three times with wash buffer (50 mM Tris HCl pH 7.5, 300 mM KCl, 1 mM MgCl_2_, 0.5% NP-40). A fourth wash was performed with ice-cold PBS and the beads were resuspended in Trizol for RNA extraction. Following Chloroform/isopropanol extraction and DNase I digestion, cDNA synthesis was performed using all RNA recovered. Finally, the samples were assessed by qPCR.

### 4.12. Northern Blot

RNA samples (20 µg) were denatured by incubating at 65 °C for 10 min in RNA loading dye (containing 45% Formamide, 1× MOPS, 2% Formaldehyde, 5% Glycerol, 0.01% Bromophenol Blue). Next, 2.3 µL of ethidium bromide (ETBr) (400 μg/μL) was added. Gel electrophoresis was performed in a 1× MOPS buffer with 1% agarose and 2% formaldehyde. Running was performed until the running front was approximately 1 cm from the gel border, with the voltage set at 70 V for the first half and 110 V for the second half of the run. The gel was then incubated at RT for 30 min each in NaOH 50 mM followed by 50 mM Tris pH 7.5 and, finally, 20× SSC. Next, RNA transfer to a nitrocellulose membrane (Amersham Hybond-N, Cytiva, Marlborough, MA, USA) was achieved by capillarity action overnight and the RNA–membrane crosslinking was performed using a UV stratalinker at 254 nm. Next, the membrane was pre-hybridized for 30 min with a hybridization buffer (containing 5× SSC, 20 mM NaPi pH 7.2, 7% SDS, 0.02% albumin fraction V, 0.02% Ficoll 400, 0.02% polyvinylpyrrolidone K30) in RNAse-free H_2_0 at 60 °C using a rotational oven. The short antisense DNA probes for *GAPDH* were labeled with γ-32P-ATP (Hartmann Analytics, Braunschweig, Germany) using T4 PNK enzyme (Life Technologies, Carlsbad, CA, USA). The cDNA probes for *DAGAR* were double-labeled internally with [α-^32^P] dCTP y [α-^32^P] dATP (Hartmann Analytics, Braunschweig, Germany). Probe labeling was performed according to the manufacturer’s instructions using the GE Healthcare Ameshram™ Megaprime™ DNA Labeling System kit (Thermo Fisher, USA). The DNA or cDNA probes were purified using Illustra MicroSpin G-25 columns (GE Healthcare, USA) and the flow-through was collected, added to the bottle, and incubated overnight. Full *DAGAR* probe sequences are shown in Appendix A. Membrane washing was performed in a turning wheel at 60 °C. The membrane hybridized with antisense DNA probes was washed twice with wash solution 1 (containing 5× SSC, 1% SDS) and once with wash solution 2 (containing 1× SSC, 1% SDS), with each wash step lasting for 10 min. For the membrane hybridized with cDNA probes, the following wash steps were performed: Wash Buffer 3 (2× SSC + 0.1% SDS) and Wash Buffer 4 (0.5× SSC + 0.1% SDS) were incubated for 15 min each, followed by Wash Buffer 5 (0.1× SSC + 0.1% SDS) for 30 min. Finally, the membrane was exposed to a phosphor screen, and the signal was scanned using a laser phospho imager (Bio-Rad Laboratories, USA).

### 4.13. N6-Methyladenosine (m6A) Immunoprecipitation from Total RNA

Total RNA from hPASMC at D0 and D2 was isolated by TRIzol (Invitrogen, Thermo Fisher, USA) according to the manufacturer’s instructions. A quantity of 10 µg of purified anti-m6A [82] (Clone 9B7) was incubated for 3 h at 4 °C with 6 µg of total RNA in 1 mL of RNA-IP buffer (50 mM Tris pH 7.5, 150 mM NaCl, 5mM EDTA, 0.5% NP-40, 10% glycerol). A quantity of 30 µL of Sepharose beads (GE Healthcare, USA) was added and preparations were incubated for 2 additional hours at 4 °C followed by three washes with washing buffer (50 mM Tris pH 7.5, 300 mM NaCl, 5 mM EDTA, 0.5% NP-40, 10% glycerol). To isolate RNA from the beads, Trizol/Chloroform/isopropanol precipitation was performed. An anti-IgG antibody was used as a control for nonspecific binding to the heavy chains of the antibody.

### 4.14. Determination of Percentage of Cells in S Phase

This was conducted by flow cytometry (Fortessa, Becton Dickinson, Franklin Lakes, NJ, USA) after propidium–iodide staining. Briefly, cells cultured in cell culture medium were permeabilized with 100% EtOH and incubated at −20 °C for 30 min. Afterwards, cells were incubated with 10 mg/mL of RNAse A and 1 mg/mL of propidium–iodide for 30 min at 37 °C in darkness.

### 4.15. Protein Extraction and Western Blot

Cells were harvested using a cell scraper and washed with phosphate-buffered saline (PBS) at 4 °C twice to remove cell debris and then incubated with lysis buffer (25 mM Tris-HCl pH 7.5, 150 mM KCl, 2 mM EDTA, 1 mM NaF, 0.5% NP-40 in H_2_O milli Q, 1 mM DTT, 500 μM AEBSF) at 4 °C for 30 min with occasional mixing. Lysate was centrifuged at 16,000 rcf for 20–45 min and the supernatant was collected. Protein concentration was calculated using Bradford Assay and extrapolating absorbance at 595 μM to the absorbance of the standard BSA curve samples. The concentration was adjusted with lysis buffer to obtain the same stock concentration for all samples. Finally, 5× Laemmli Buffer was added and incubated at 95 °C for 5 min. At this point the samples were used for gel electrophoresis or stored at −20 °C.

Electrophoresis was performed on 10% sodium dodecyl sulfate polyacrylamide gel (SDS-PAGE) with standard running buffer. The transfer was carried out using Towbin buffer (25 mM Tris, 192 mM glycine, 20% (*v*/*v*) methanol (pH 8.3)) in a semi-dry transfer chamber (Trans-Blot SD Semi-Dry Transfer Cell, Bio-Rad, USA) at constant intensity (1 min per kDa at 2 mA per cm^2^ of gel surface). Amersham Protran Premium 0.45 μm nitrocellulose membranes (Cytiva, USA) were used. Blocking was performed by 30 min incubation with 0.1% TBS-Tween + 2.5% bovine milk while shaking. Primary antibodies were incubated overnight at 4 °C in motion in 0.1% TBS-Tween + 1% bovine milk and Li-Cor-suitable secondary antibodies were incubated for 1 h in 0.1% TBS-Tween + 1% bovine milk in motion and protected from light at RT. Glyceraldehyde-3-phosphate-dehydrogenase (GAPDH) was used as a loading control. Scanning of the membranes was performed on a LI-COR reader (Li-Cor Biosciences, Lincoln, NE, USA). Primary antibodies used included anti-YTHDF2 (Helmholtz Zentrum München and Meister Lab, Oberschleißheim, Germany, private antibody, rat monoclonal, clone 9G11), anti-YTHDF2 (Proteintech Group Inc., Fisher Scientific, Rosemont, IL, USA, rabbit polyclonal, 24744-1-AP), and anti-GAPDH (GeneTex, Irvine, CA, USA, mouse monoclonal, GTX627408P, GT239). Secondary antibodies used included anti-rat IgG 680 (Li-Cor Bioscience, Lincoln, NE, USA, goat monoclonal, 925-68076), anti-rat IgG 800 (Li-Cor Bioscience, Lincoln, NE, USA, goat monoclonal, 926-32219), anti-rabbit IgG 680 (Li-Cor Bioscience, Lincoln, NE, USA, goat monoclonal, 926-32221), anti-rabbit IgG 800 (Li-Cor Bioscience, Lincoln, NE, USA, goat monoclonal, 926-32211) and anti-mouse IgG 680 (Li-Cor Bioscience, Lincoln, NE, USA, goat monoclonal, 926-32220).

### 4.16. Statistical Analysis

Statistical analysis was conducted to meet the requirements of the respective models. Those corresponding to one factor and two treatments were analyzed using either regular or one-sample Student *t*-test (two-tailed) unless otherwise indicated. For comparisons involving three or more groups, a one-way ANOVA was performed, followed by post hoc pairwise comparisons using Tukey tests. All *p*-values < 0.05 were considered significant. RT-qPCR data were analyzed using the Log(2) transformation of the normalized relative expression as suggested by [83] and graphs were made using the untransformed variable for intuitive visualization.

## Figures and Tables

**Figure 1 ijms-25-09497-f001:**
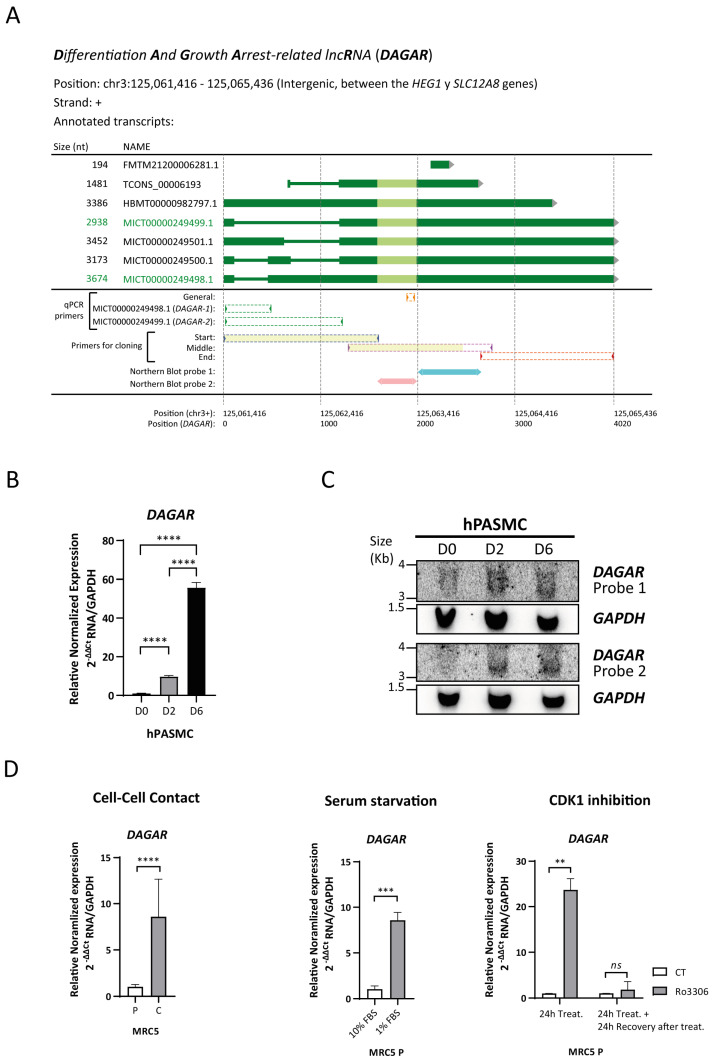
*DAGAR* characterization. (**A**) Scheme of *DAGAR* locus and annotated variants: qPCR primers and Northern blot probes are aligned below. Light green depicts the area with greater conservation between species. In yellow are depicted the regions obtained and sequenced with high confidence in cloning experiments. MICT00000249498.1 corresponds to *DAGAR-1* and MICT00000249499.1 to *DAGAR-2*. nt = nucleotides; qPCR = quantitative Polymerase Chain Reaction (**B**) Expression of *DAGAR* during hPASMC differentiation by RT-qPCR (*n* = 3). Data were analyzed using two-tailed one-way ANOVA on Log(2) transformation of relative normalized expression. *p* values displayed account for Tukey multiple-comparison post hoc test. (**C**) Northern blot of *DAGAR*, with *GAPDH* as loading control. (**D**) *DAGAR* expression in cell-cycle-arrested fibroblast (left panel) after induction of quiescence by cell-to-cell contact (*n* = 5; data were analyzed using two-tailed Student *t*-test on Log(2) transformation of relative normalized expression), (middle panel) in proliferative MRC5 cells after serum starvation for 48 h compared with normal growth medium (*n* = 3, analyzed using a two-tailed Student *t*-test on Log(2) transformation of relative normalized expression), and (right panel) after reversible CDK1 inhibition with 10 μM of Ro3306 treatment for 24 h (*n* = 2) and after additional 24 h of recovery following inhibitor withdrawal (*n* = 2, analyzed using a two-tailed Student *t*-test with Welch’s correction on Log(2) transformation of relative normalized expression). Same volume of DMSO was used as vehicle control for Ro3306 treatment. MRC5 P = cells at 70% confluency. MRC5 C = cells 48 h after reaching full confluency. hPASMC = Human Pulmonary Artery Smooth Muscle Cell. D0 = Cells at 70% Confluency. D2 = Cells at full confluency. D6 = Cells 96 h after reaching full confluency. ** *p* < 0.01, *** *p* < 0.001, and **** *p* < 0.0001.

**Figure 2 ijms-25-09497-f002:**
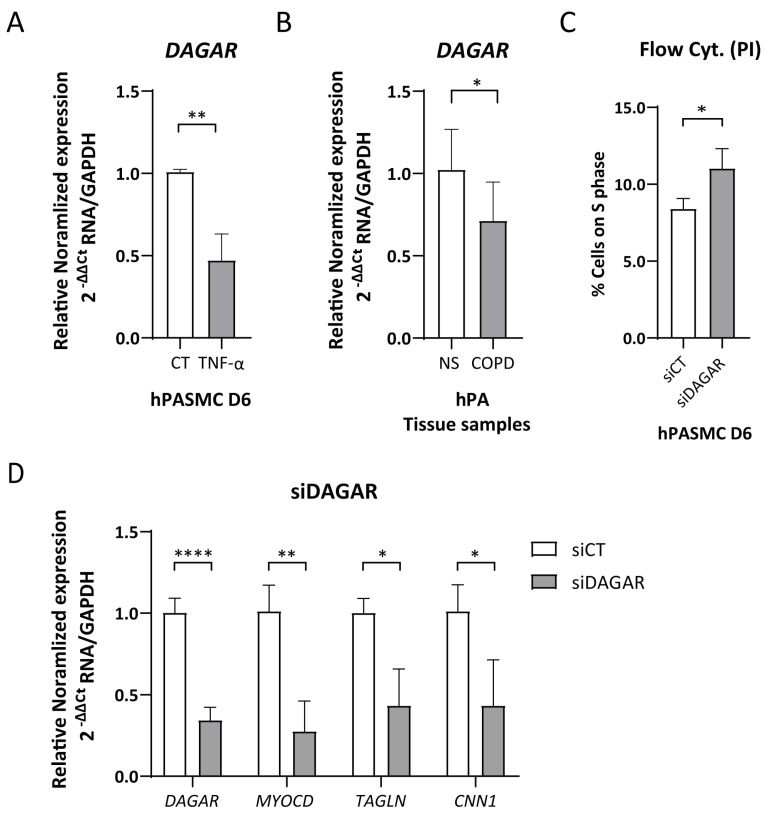
*DAGAR* expression regulation due to pro-inflammatory signaling and in COPD patients. (**A**) *DAGAR* expression after 48 h of 10 ηg/μL of TNFα treatment compared to vehicle analyzed by RT-qPCR (*n* = 6, data were analyzed using Mann–Whitney test on Log(2) transformation of relative normalized expression). (**B**) *DAGAR* expression in total RNA from pulmonary arteries derived from COPD patients compared to healthy non-smokers. COPD = Chronic Obstructive Pulmonary Disease (*n* = 8); NS = Non-Smokers (*n* = 5). Data were analyzed using a two-tailed Student *t*-test on Log(2) transformation of relative normalized expression. (**C**) Percentage of cells on S phase (*n* = 4; data were analyzed using a two-tailed Student *t*-test). (**D**) Effects of *DAGAR* knockdown after 48 h of siPOOL transfection on hPASMC differentiation. *DAGAR* and SMC marker gene expression were analyzed by RT-qPCR with *GAPDH* as loading control. Relative mRNA levels in siDAGAR compared to siCT (scrambled) normalized to *GAPDH* (*n* = 5; data were analyzed using two-tailed Student *t*-test for *DAGAR* and Welch’s *t*-test for MYOCD, CNN1, and Sm22a on Log(2) transformation of normalized relative expression) * *p* < 0.05, ** *p* < 0.01, and **** *p* < 0.0001.

**Figure 3 ijms-25-09497-f003:**
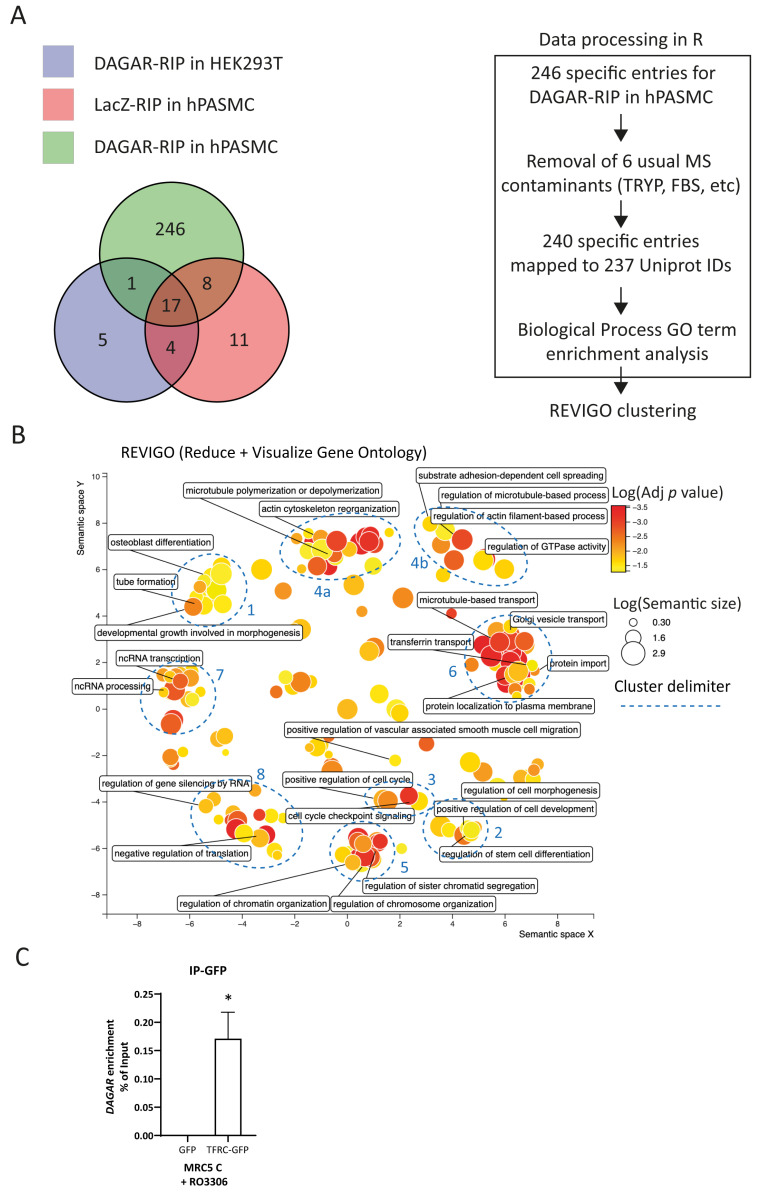
Mass spectrometry analysis of *DAGAR* RNA pulldown (DAGAR-RIP). (**A**) Left panel: Venn diagram on proteins identified from mass spectrometry data on control samples (LacZ-RIP on hPASMCs and DAGAR-RIP on HEK293T cells) and specific samples (DAGAR-RIP on hPASMCs). Right panel: Data curation, Uniprot mapping, and Gene Ontology enrichment analysis for biological processes using clusterProfiler within the R environment followed by semantic reduction using REVIGO platform. Additionally, Reactome Pathway enrichment analysis was performed (Appendix A) using ReactomePA within the R environment. (**B**) REVIGO semantic space reduction of significantly enriched Gene Ontology terms of biological processes and manual cluster delimitation (blue). (**C**) TFRC-*DAGAR* interaction cross-validation on confluent MRC5 cells transfected with plasmid expressing either GFP-TFRC or GFPalone and induced for 24 h with 10 μM of Ro3306 for maximum *DAGAR* expression. GFP-pulldown using an anti-GFP followed by RT-qPCR (*n* = 3; data were analyzed using one sample *t*-test on *DAGAR* percentage of input recovery after pulldown). * *p* < 0.05.

**Figure 4 ijms-25-09497-f004:**
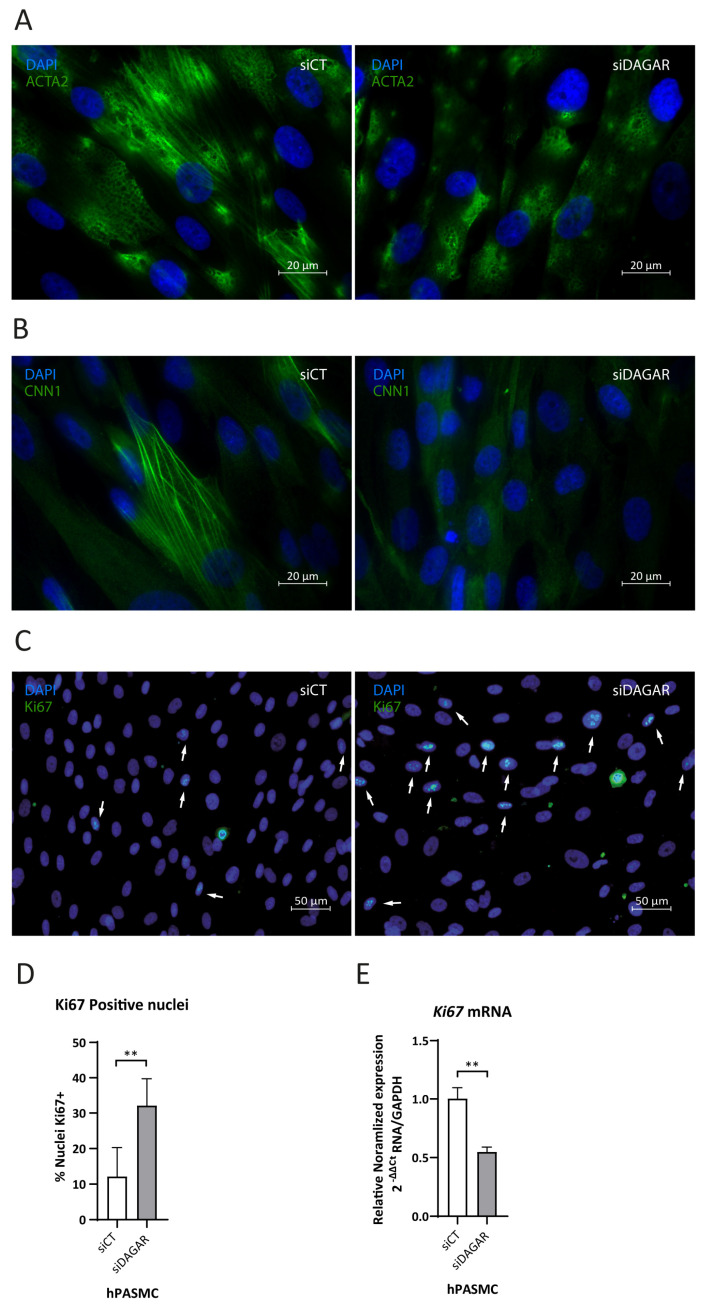
Effects of *DAGAR* knockdown after 48 h of siPOOL transfection on hPASMC cytoskeleton formation and ki67 nuclear localization by immunofluorescence. (**A**) α-SMA fiber formation in control (scrambled) siPOOL compared with siDAGAR. (**B**) CNN1 fiber formation in control (scrambled) siPOOL compared with siDAGAR. (**C**) Ki67 nuclear localization; white arrows point to DAPI-Ki67 colocalization in control (scrambled) siPOOL compared with siDAGAR. (**D**) Percentages of Ki67 + cells (*n* = 6; data were analyzed using two-tailed Student *t*-test). (**E**) RNA expression of Ki67 analyzed by RT-qPCR (*n* = 3; data were analyzed Welch’s *t*-test on log(2) transformation of the normalized relative expression). ** *p* < 0.01.

**Figure 5 ijms-25-09497-f005:**
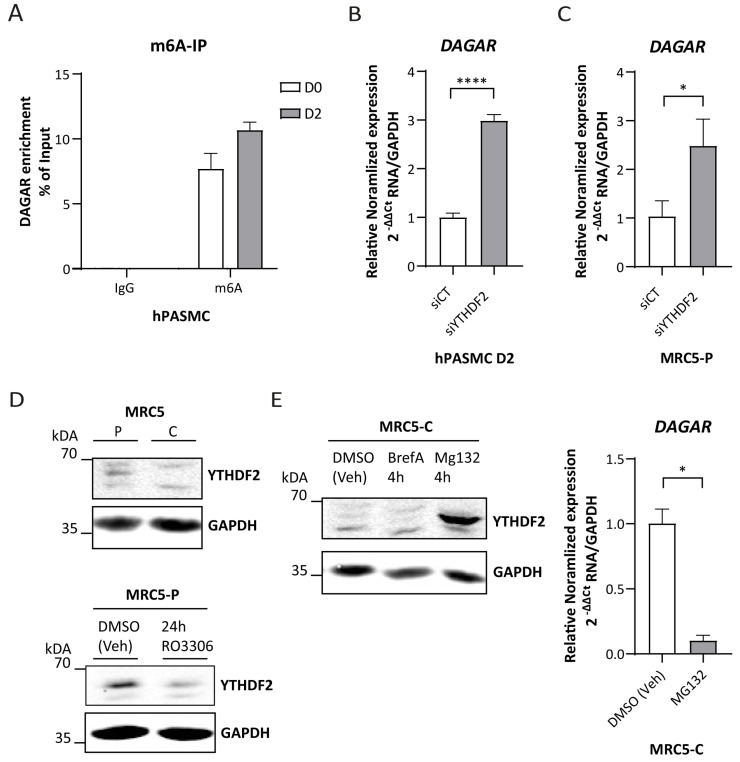
*DAGAR* regulation by the YTHDF2/m6A axis. (**A**) m6A immunoprecipitation from total RNA derived from hPASMCs at D0 and D2 with anti-IgG as control. (**B**) *DAGAR* expression after silencing YTHDF2 m6A reader (siYTHDF2) compared to scrambled siPOOL control (siCT) in hPASMC measured by RT-qPCR (*n* = 3; data were analyzed using two-tailed Student *t*-test on log(2) transformation of the normalized relative expression). (**C**) *DAGAR* expression after silencing YTHDF2 reader (siYTHDF2) compared to scrambled siPOOL control (siCT) in proliferative MRC5 cells measured by RT-qPCR (*n* = 3; data were analyzed using two-tailed Student *t*-test on log(2) transformation of the normalized relative expression). (**D**) Western blot of YTHDF2 in confluent MRC5 cells (MRC5 C) compared to proliferative MRC5 cells (MRC5 P) (upper panel); Western blot of YTHDF2 after Ro3306-induced cell cycle exit in proliferative MRC5 cells (MRC5-P 24 h Ro3306) compared to DMSO (vehicle)-treated proliferative MRC5 cells (MRC5-P DMSO veh) (lower panel). GAPDH was used as loading control. (**E**) *DAGAR* expression after YTHDF2 recovery. Western blot of YTHDF2 after proteasome inhibition using MG132 treatment (MG132 4 h); retrograde transport inhibition by Brefeldin A treatment (BrefA 4 h) or DMSO (vehicle) treatment (DMSO veh) (right panel); *DAGAR* expression measured by RT-qPCR in MRC5 confluent cells after proteasome inhibition using MG132 treatment compared to DMSO vehicle control condition (left panel) (*n* = 2, analyzed using Student *t*-test on log(2) transformation of normalized relative expression). * *p* < 0.05, **** *p* < 0.0001.

## Data Availability

All datasets generated have been deposited on public repositories and will be available at the time of publication. Accession Numbers: RNAseq data: GEO accession number: GSE242196; Masspec data: PRIDE access to the data is under project accession PXD029550.

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
