# Peer review of "Differentiation and Growth-Arrest-Related lncRNA (*DAGAR*): Initial Characterization in Human Smooth Muscle and Fibroblast Cells"

_ijms, 2024, doi:10.3390/ijms25179497_

Round 1

Reviewer 1 Report

Comments and Suggestions for Authors

This manuscript attempts to unravel factors that account for the dedifferentiation change in proliferation status of SMC and their behaviour in such disorders as COPD. A novel lncRNA DAGAR is described whose altered expression might be related to this change in growth pattern owing to its interaction with cell cycle related factors. Some solid work has been done in support of the hypothesis and the authors do not overinterpret their findings. However, there are some major caveats that need addressing. There are no ladders or molecular weight markers described for the blot experiments which makes their interpretation somewhat unclear. (Whilst there are markers given beside the letter box figures there is no evidence to support their positioning). The probes for the Northern blot are not given. In addition there is no data for the RNA quality. RNA concentration has been measure using the Nanodrop but there is no information concerning the actual RNA quality. Maybe the ethidium bromide stained gel or a Bioanalyser trace would demonstrate that RNA of good quality with strong rRNA bands had been produced. Otherwise there remains the possibility that some of the result have arisen from differential RNA quality. (RNA concentration is simply an absorbance reading at 260nm). The immunofluorescence figures are not entirely convincing and the scale bars not fully labelled.

The manuscript has definite merit but as yet the presentation is not sufficient for all to be believed.

Author Response

We thank the reviewer for taking the time to read and make interesting comments on our work.

Comment 1. There are no ladders or molecular weight markers described for the blot experiments which makes their interpretation somewhat unclear. (Whilst there are markers given beside the letter box figures there is no evidence to support their positioning).

Response 1. We thank the reviewer for pointing this out. All Western Blots are provided with the original blot as supplementary information. Ladders are now labeled with their band sizes to provide clarity for the reviewer. The ladder used was the PAGE-ruler pre-stained (from 10 to 180 kDA). For Northern Blots, Ladder was run in the gel, but is not stained in the final membrane, therefore size estimation was made by superposing the ethidium bromide gel staining with the signal from the membrane. (Please find attached the images of the gels and labeled ladder)

Comment 2. The probes for the Northern blot are not given.

Response 2. We thank the reviewer for the comment, and although the Northern Blot probes are schematized in Figure 1 and the primers used to make the probes provided in supplementary table 1 (primers), we have not provided the full-length sequence. We have noticed that the name of the probes in the figure 1 was inverted. Please find attached the corrected Figure 1 and Supplementary table 1. Additionally, we have added the full-length sequence of the probes used in supplementary table 1 and indicated in lines 699 and 700 of the material and methods section.

Comment 3. In addition there is no data for the RNA quality. RNA concentration has been measure using the Nanodrop but there is no information concerning the actual RNA quality. Maybe the ethidium bromide stained gel or a Bioanalyser trace would demonstrate that RNA of good quality with strong rRNA bands had been produced. Otherwise there remains the possibility that some of the results have arisen from differential RNA quality. (RNA concentration is simply an absorbance reading at 260nm).

Response 3. We thank the careful review of the data provided. Indeed, RNA concentration measured by NanoDrop does not give any information regarding the integrity of the extracted RNA. We are happy to contribute with clarifying data and statements regarding the RNA quality for the Northern Blot presented for characterization, as well as for siPOOL experiments. Of note, all these experiments were performed within an RNA biology lab (Biochemistry I department, Regensubrg University, Regensburg, Germany), where all the precautions for RNA extraction and downstream experiments are met. In fact, since Northern Blot sample preparation involves a denaturing step heating the sample to 65°C for 10 minutes in RNA loading dye (containing 45% Formamide, 1x MOPS, 2% Formaldehyde, 5% Glycerol, 0.01% Bromophenol Blue), any type of RNAse contamination will quickly degrade the sample and turn yellow the bromophenol blue. RNA extraction is performed using only RNAse and DNAse clean sterile (autoclaved) material (i.e. filtered pipette tips, Eppendorf tubes, Falcon tubes) and RNAse decontaminated surfaces. Samples were kept on ice throughout the whole process and RNA quality was assessed prior to NB applications, since as mentioned, minimal contamination can destroy the sample. Nevertheless, for qPCR experiments, all the same precautions mentioned were used, with the difference that extraction was performed using a silica-membrane based kit, which diminishes greatly the chance of degradation and DNA contamination in comparison to Trizol-based extraction methods (which was used for Northern Blot applications due to higher yield capabilities of this type of extraction compared to columns). RNA extraction and cDNA retrotranscription were performed within the same day, in some cases extending even to the qPCR measurements (although not in all cases), in which case cDNA was stored at -20°C until qPCR was performed. All RNA extracted was kept on ice and stored at -80°C. As an additional caution, cDNA was prepared using random primers, which will fairly represent RNA even when some degradation has occurred, given the fact that RNA starts degrading from the polyadenylated tail of the transcript when RNAses are active. This contrasts with the use of Oligo(dT) primers, which bind to the poli A tail of RNAs to start retrotranscription and can underestimate the presence of partially degraded RNAs present in the sample.

Comment 4. The immunofluorescence figures are not entirely convincing and the scale bars not fully labelled.

Response 4. We thank the reviewer for the comment and corrections have been made to the scale bars. Regarding the immunofluorescences presented, please note the lack of fibers forming on siDAGAR condition when compared to siCT (scrambled) for both alpha actin (ACTA2) and calponin1 (CNN1). We do not state that there are changes in staining intensity (although for CNN1 there is an evident general decrease), but rather a morphological change (see lines 281-283 of the results section and in line 411 of discussion section). We have selected the best representative images to show in the manuscript. A new control image was selected for the staining of CNN1. In addition, we send additional images for the reviewer to show the consistency of the observed phenotype. Please let us know if the reviewer wants us to add more images or change the selected ones. Moreover, if requested we are willing to send the original images.

For a more aesthetical view of the Figure, we have moved IF quantification to Figure 4D and added Figure 4E for mRNA quantification (310-313) and stated in the text (line 287).  

Reviewer 2 Report

Comments and Suggestions for Authors

This manuscript focused on long non-coding RNAs (IncRNAs) as one of the key modulators of vascular smooth muscle cells (SMC), and their role in the development of diseases such as Chronic Obstructive Pulmonary Disease (COPD). The authors discovered a related lncRNAs, which they named it DAGAR. By using bioinformatics methods, they demonstrated that DAGAR is upregulated in quiescent

with respect to proliferative SMC and in cell cycle arrested MRC5 lung fibroblasts.

Some issues need to be fixed:

1.     Lines 130—132 are irrelevant to this study.

2.     There are two subheadings of Section 2.3.

3.     Some subheadings are in bold while some are not.

4.     Lines 281—291 are not aligned with other sections.

5.     Fig. 3A shows a Venn diagram. But why not using a standard 3-circular Venn diagram? Current one looks weird.

Author Response

We thank the reviewer for their constructive comments. We have worked to improve the manuscript according to reviewers suggestions.  

Comment 1. Lines 130—132 are irrelevant to this study.

Response 1. We thank the reviewer for his comment. The lines 130-132 have been deleted.

Comment 2. There are two subheadings of Section 2.3.

Response 2. The subheadings have been corrected.

Comment 3. Some subheadings are in bold while some are not.

Response 3. We apologize for the inconsistence. The subheadings have been corrected accordingly.

Comment 4. Lines 281—291 are not aligned with other sections.

Response 4. We thank the reviewer noticing it. We have now aligned the lines according to the other sections.

Comment 5. 3A shows a Venn diagram. But why not using a standard 3-circular Venn diagram? Current one looks weird.

Response 5. We have now changed figure 3 including a standard 3-circular Venn diagram.

Round 2

Reviewer 1 Report

Comments and Suggestions for Authors

I thank the authors for resolving my reservations from the original manuscript and my concerns have been answered.